# Numerical Determination of the Frictional Coefficients of a Fluid Film Journal Bearing Considering the Elastohydrodynamic Lubrication and the Asperity Contact Force

**Gwanghee Hong, Kyobong Kim, Youngjun Park and Gunhee Jang ***

Precision Rotating Electromechanical Machine Laboratory, Department of Mechanical Convergence Engineering, Hanyang University, Seoul 04763, Korea; rhkdgml125@naver.com (G.H.); kkb4477@naver.com (K.K.); lkjk0418@naver.com (Y.P.)
* Correspondence: ghjang@hanyang.ac.kr

**Abstract:** This paper proposes a numerical method to investigate the frictional characteristics of a fluid film journal bearing considering the elastohydrodynamic lubrication and the asperity contact force. We solved the average Reynolds equation by utilizing the FEM to determine the hydrodynamic force developed by the lubricant of the journal bearing. We also used a modified GT model (Greenwood–Tripp model) developed by Greenwood and Tripp to decompose the asperity contact force into normal and tangential directions. Once we applied those forces to a rotor, we solved the equations of motion of a flexible shaft to determine the friction coefficient. We verified the proposed method by comparing the calculated friction coefficient with the measured one of journal bearings conducted by prior researchers. Then, the proposed method was applied to investigate the frictional characteristics of a journal bearing of a scroll compressor on which dynamic loads are applied. This paper can contribute to developing robust rotor systems supported by journal bearings.

**Keywords:** asperity contact force; elastohydrodynamic lubrication; journal bearing; Stribeck curve

## 1. Introduction

Fluid film journal bearings support a rotor through pressure developed in lubricant, but a contact of the shaft with the bearing occurs when rotating speed is low or a large external force is applied. The contact in a mechanical device causes wear, which is one of the major causes of shortening life and jeopardizing the operation of a rotor-bearing system. Analysis of the contact phenomena and friction characteristics is essential to enhance the lifespan in a rotor and bearing system, and the Stribeck curve is a guideline to analyze these frictional characteristics in fluid bearings.

The Stribeck curve is a graph of the friction coefficients according to the Hersey number [1]. The Hersey number is a dimensionless number of rotational velocity, external force, and viscosity of the lubricant. It divides the frictional properties of a rotor and bearing system into three lubricating regions of hydrodynamic, mixed, and boundary lubrications. Figure 1 is an example of the Stribeck curve, and the three lubricating regions can be identified as the boundary lubrication with solid and solid contact, mixed lubrication with solid and fluid contact, and hydrodynamic lubrication supported only by lubricants. Based on the interval in which the friction coefficient of the derived graph changes rapidly, the lubricating regions can be distinguished, and the rotation speed and external force in each lubricating region can be determined. A method of analyzing this friction characteristic is an important technique to predict the degree of wear generated by a rotating machine [2–4]. Since abrasion caused by friction during the steady state operation is fatal to the lifespan of the rotating device, predicting friction characteristics of a rotor and bearing system has been steadily required in the field of tribology.

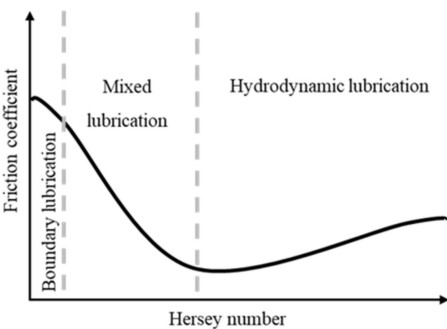

**Figure 1.** Classification of lubrication regimes of fluid film journal bearings.

Previous researchers conducted experiments to determine frictional coefficients in the Stribeck curve under various conditions. X. Lu et al. [5,6] used two types of lubricants to obtain the Stribeck curve by considering oil temperature. Boncheol Ku et al. [7] constructed the experimental equipment with PETE (polytetrafluoroethylene) compounds and aluminum alloys on the surface of the journal bearing to investigate lubrication characteristics on two materials. Xin Ai Zhang et al. [8] measured the friction coefficients for various oils with a pin-in-disk tribometer and summarized them in the form of Stribeck curves. Jianqiao Hu et al. [9] studied surface friction at the nanoscale level and performed frictional tests.

Several papers have been proposed to analytically obtain frictional properties without experiments. E. R. M. Gelink et al. [10] calculated the friction coefficient of mixed lubrication by utilizing the surface roughness obtained through experiments. Alex de Kraker et al. [11] used the EHL (Elastohydrodynamic Lubrication) theory to derive the Stribeck curves considering elastic lubrication in very small oil film thickness. Furthermore, A. Cubillo et al. [12] analyzed the oil film pressure of journal bearings using TEHL (Thermo-Elastohydrodynamic Lubrication) theory, which considered the viscosity of lubricants to vary with temperature. Zhuming Bi et al. [13] proposed a method of setting up an EHL model and how to verify it. Tao He et al. [14] used a new approximation formula based on a modified Dyson model to derive the Stribeck curve with rotation speed.

However, the prior studies derived the Stribeck curve by measuring the friction coefficient between the rotating and the stationary parts on which the static load was applied. Prior researchers could not determine the Stribeck curve analytically without relying on experiments. In this study, we solved the average Reynolds equation considering elastohydrodynamic lubrication (EHL) by utilizing the FEM to determine bearing force and friction force developed by the lubricant. We also used a modified GT model (Greenwood–Tripp model) developed by Greenwood and Tripp to decompose the asperity contact force into normal and tangential directions. Once we applied those forces to a rotor, we solved the equations of motion to determine the friction coefficient. Then, the proposed method was applied to investigate the frictional coefficients for the rotating shaft of a SC (scroll compressor) on which dynamic loads are applied. Additionally, the bearing loss and contact loss were calculated and investigated.

## 2. Numerical Calculation Methods of the Frictional Coefficients and Power Loss

### 2.1. Reynolds Equation Considering EHL

The Reynolds equation of a journal bearing in cylindrical coordinates can be described as follows [15]:

$$\frac{\partial}{R\partial\theta}\left(\frac{h^3}{12\mu}\frac{\partial P}{R\partial\theta}\right) + \frac{\partial}{\partial z}\left(\frac{h^3}{12\mu}\frac{\partial P}{\partial z}\right) = \tilde{U}_e\frac{\partial h}{R\partial\theta} + \frac{\partial h}{\partial t} \tag{1}$$

where $R$, $h$, $\mu$, $P$, and $\tilde{U}_e$ represent the shaft radius, the fluid film thickness of the journal bearing, the fluid viscosity, the pressure of oil film, and the average lubricant velocity, respectively. Pressure of journal bearing can be obtained by solving this governing equation. Recently, in the area where the fluid film thickness between the rotating and the stationary

parts is ultra-thin in which lubricants are affected by surface roughness, the governing equation of bearing pressure considering this phenomenon is proposed by adding the effect of pressure change due to asperities on the surface to the conventional Reynolds equations. This governing equation is called an average Reynolds equation, which can calculate the pressure of oil film by considering EHL [16]. EHL is a lubrication in which elastic deformation of the asperities on surfaces takes place to affect lubricant film thickness in the contact. The average Reynolds equation of a journal bearing in cylindrical coordinates can be described as follows:

$$\frac{\partial}{R\partial\theta}\left(\phi_\theta\frac{h^3}{12\mu}\frac{\partial P}{R\partial\theta}\right) + \frac{\partial}{\partial z}\left(\phi_z\frac{h^3}{12\mu}\frac{\partial P}{\partial z}\right) = \phi_c\left[\tilde{U}_e\frac{\partial h}{R\partial\theta} + \frac{\partial h}{\partial t}\right] + \tilde{U}_e\sigma\frac{\partial\phi_s}{R\partial\theta} \tag{2}$$

where $\sigma$ means the standard deviation of the height of surface asperities, and it is calculated using the rms average of the standard deviations of two surfaces [17]. Here, $\phi_x$ and $\phi_y$ are the pressure flow factors according to the shape of the asperities on each surface, $\phi_c$ is the contact factor, and $\phi_s$ is the shear factor determined by the standard deviation of the surface asperities and the oil film clearance. In this study, the oil film pressure generated by the rotor and bearing system is calculated by FEM. The finite element equation of the average Reynolds equation can be written as follows:

$$\left[\iint_A \phi_\theta\frac{h^3}{12\mu}\left(\frac{\partial\mathbf{N}}{R\partial\theta}\frac{\partial\mathbf{N}^T}{R\partial\theta} + \frac{\partial\mathbf{N}}{\partial z}\frac{\partial\mathbf{N}^T}{\partial z}\right)dA\right]\mathbf{p}$$
$$= \iint_A \phi_c\left\{h\left(\tilde{U}_{e,\theta}\frac{\partial\mathbf{N}}{R\partial\theta} + \tilde{U}_{e,z}\frac{\partial\mathbf{N}}{\partial z}\right) - \mathbf{N}\frac{\partial h}{\partial t}\right\}dA + \iint_A \phi_s\sigma\left(\tilde{U}_{e,\theta}\frac{\partial\mathbf{N}}{R\partial\theta} + \tilde{U}_{e,z}\frac{\partial\mathbf{N}}{\partial z}\right)dA \tag{3}$$

where $\mathbf{N}$, $\mathbf{p}$, $A$ are the vector of the shape function of the bilinear element with four nodes, pressure of the element, and an element area. $\tilde{U}_{e,\theta}$ and $\tilde{U}_{e,z}$ are the average lubricant velocities of the rotational and axial directions, respectively. The pressure of journal bearing considering surface roughness can be obtained by solving the global matrix equation constructed by the local matrix in Equation (3). Boundary conditions for the journal bearing is the half Sommerfeld condition in the circumferential direction and the ambient pressure in both sides. The bearing force generated by the pressure of oil film can be expressed in the following equation:

$$F_{B,N} = \int\int Pd\theta dz \tag{4}$$

In the same way, shear stress can be used to calculate the friction force generated by the lubricant. The frictional force can be expressed in the following equation:

$$F_{B,T} = \int\int\left(\frac{h}{2}\frac{\partial P}{R\partial\theta} + \mu\frac{R\dot{\theta}}{h}\right)d\theta dz \tag{5}$$

where $\dot{\theta}$ is the rotating speed. Friction torque caused by the lubricant can be obtained by multiplying Equation (5) by the bearing radius.

### 2.2. Modified Asperity Contact Model Considering Tangential Direction

Greenwood's asperity contact theory is used to determine the contact force when two surfaces contact each other. Greenwood assumed that many small hemispherical asperities exist on the surface and that the asperities on each surface follow a Gaussian distribution. The force generated by one asperity is as follows [18]:

$$f_{contact} = \frac{4}{3}E'\beta^{1/2}\left(z - h - \frac{r^2}{2\beta_s}\right)^{3/2} \tag{6}$$

where $E'$, $\beta$, $r$, and $\beta_s$ mean the composite elastic modulus of the rotating shaft and bearing, the rms value of the surface radius of asperity, the distance between the asperities, and the

sum of the two asperities radii, respectively. The composite elastic modulus E' considers the elastic modulus of the two objects in contact. It can be obtained with the following equation:

$$\frac{1}{E'} = \frac{1 - \varphi_1{}^2}{E_1} + \frac{1 - \varphi_2{}^2}{E_2} \tag{7}$$

where $\varphi_1$ and $\varphi_2$ are the Poisson's ratio of the two objects in contact, respectively. Figure 2 shows the normal and tangential components of the contact forces.

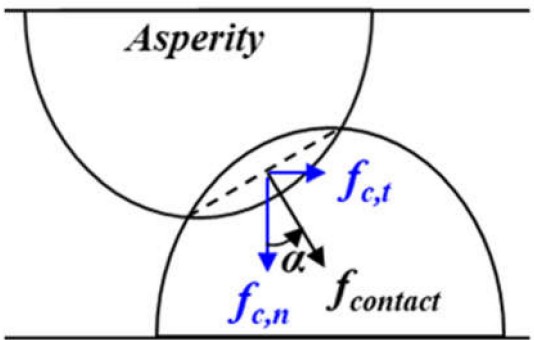

**Figure 2.** Contact force between two asperities.

The contact forces between the asperities of two surfaces are determined into normal and tangential directions as follows:

$$f_{c,n} = f_{contact} \cos \alpha = \frac{4}{3} E' \beta^{1/2} \left( z - h - \frac{r^2}{2\beta_s} \right)^{3/2} \left( 1 + \frac{r^2}{\beta_s{}^2} \right)^{-1/2} \tag{8}$$

$$f_{c,t} = f_{contact} \sin \alpha = \frac{4}{3} E' \beta^{1/2} \left( z - h - \frac{r^2}{2\beta_s} \right)^{3/2} \left( 1 + \frac{r^2}{\beta_s{}^2} \right)^{-1/2} \frac{r}{\beta_s} \tag{9}$$

where $\alpha$ is the angle between the asperities of two surfaces. Total contact forces in normal and tangential directions are obtained by integrating Equations (8) and (9) with respect to the contact area as follows:

$$F_{C,N} = \frac{8}{3\sqrt{2\pi}} \pi A E' \eta_1 \eta_2 \beta^{1/2} \sigma^4 \int_h^\infty \int_0^{\sqrt{2\beta_s(z-h)}} \left( z - h - \frac{r^2}{2\beta_s} \right)^{3/2} \left( 1 + \frac{r^2}{\beta_s} \right)^{-1/2} e^{-z^2/2} r \, dr \, dz \tag{10}$$

$$F_{C,T} = \frac{8}{3\sqrt{2\pi}} A E' \eta_1 \eta_2 \frac{\beta^{1/2}}{\beta_s} \sigma^4 \int_h^\infty \int_0^{\sqrt{2\beta_s(z-h)}} \left( z - h - \frac{r^2}{2\beta_s} \right)^{3/2} \left( 1 + \frac{r^2}{\beta_s} \right)^{-1/2} e^{-z^2/2} r^2 \, dr \, dz \tag{11}$$

where $A$, $\eta_1$, and $\eta_2$ are the contact area, the asperity density of the rotor, and the bearing, respectively. Equations (10) and (11) can be used to calculate $F_{C,N}$ and $F_{C,T}$ generated by the contact.

### 2.3. Friction Coefficient and Power Loss of a Journal Bearing

The nature of the friction loss depends on the lubrication regime. For thick fluid films, the friction is basically caused by the lubricant viscous effects. At mixed lubrication regimes a mixed friction can be observed, associated with the lubricant viscous effects and the solid contact. Once we determine the bearing and friction forces of the lubricant as shown in Equations (3) and (4) and the contact forces by the contact mechanism as shown in Equations (10) and (11), we can determine the friction coefficient as follows:

$$\mu_F = \frac{Tangential\ force}{Normal\ force} = \frac{F_{C,T} + F_{B,T}}{F_{C,N} + F_{B,N}} \tag{12}$$

The power loss is the loss that occurs between the rotor and the bearing when a shaft rotates with respect to the bearing. The first one is the hydrodynamic frictional loss caused by the viscosity of the lubricant in thick fluid films. The second one is the contact friction loss caused by the asperity contact when the shaft contacts the bearing in a low-speed range or in the application of a large external force. They can be represented as follows:

$$Power\ loss_B = F_{B,T}\omega R \tag{13}$$

$$Power\ loss_{Contact} = F_{C,T}\omega R \tag{14}$$

The proposed method to calculate power loss makes it possible to compare the power loss generated by contact with the power loss by lubricants in various operating conditions of a rotor and bearing system.

### 2.4. Analysis Method of Shaft Dynamics

Shaft dynamics in a SC can be determined by solving the equation of motion including external, hydrodynamic, and contact forces. The rotating shaft is modeled as a Timoshenko beam without including rotary inertia and gyroscopic effects because they are not significant in the model of this paper, and the FEM is applied to the Timoshenko beam with the consideration of bending in the $xz$ and $yz$ axes. Figure 3 shows the coordinates of a beam element in $xz$ and $yz$ planes. The displacement vector and local mass and stiffness matrices for one beam element with two nodes can be represented as follows [19]:

$$\mathbf{u} = \begin{bmatrix} U_1 & V_1 & \theta_{x1} & \theta_{y1} & U_2 & V_2 & \theta_{x2} & \theta_{y2} \end{bmatrix}^{\mathbf{T}} \tag{15}$$

$$\mathbf{M} = \begin{bmatrix} M_1 + M_7 & 0 & 0 & M_2 + M_8 & M_3 - M_7 & 0 & 0 & M_4 + M_8 \\ & M_1 + M_7 & -M_2 - M_8 & 0 & 0 & M_3 - M_7 & -M_4 - M_8 & 0 \\ & & M_5 + M_9 & 0 & 0 & M_4 + M_8 & M_6 + M_{10} & 0 \\ & & & M_5 + M_9 & -M_4 - M_8 & 0 & 0 & M_6 + M_{10} \\ & & & & M_1 + M_7 & 0 & 0 & -M_2 - M_8 \\ & & & & & M_1 + M_7 & M_2 + M_8 & 0 \\ & & & & & & M_5 + M_9 & 0 \\ Sym. & & & & & & & M_5 + M_9 \end{bmatrix} \tag{16}$$

$$\mathbf{K} = \begin{bmatrix} K_1 & 0 & 0 & K_2 & -K_1 & 0 & 0 & K_2 \\ & K_1 & -K_2 & 0 & 0 & -K_1 & -K_2 & 0 \\ & & K_3 & 0 & 0 & K_2 & K_4 & 0 \\ & & & K_3 & -K_2 & 0 & 0 & K_4 \\ & & & & K_1 & 0 & 0 & -K_2 \\ & & & & & K_1 & K_2 & 0 \\ & & & & & & K_3 & 0 \\ Sym. & & & & & & & K_3 \end{bmatrix} \tag{17}$$

where $\rho$ is the density, $\lambda$ is $(12EI)/(\kappa GAL^3)$, $E$, $I$, $\kappa$, $G$, $A$, and $L$ are the elastic modulus, the moment of inertia, the shape coefficient, the shear coefficient, the area, and the beam element length, respectively. We assemble the element matrices into a global matrix as follows:

$$\mathbf{M}\ddot{\mathbf{u}}_t + \mathbf{K}\mathbf{u}_t = [\mathrm{F}_{External} + \mathrm{F}_{Bearing} + \mathrm{F}_{Contact}]_t \tag{18}$$

where M and K are the mass and stiffness matrices, respectively, and $\mathrm{F}_{External}$, $\mathrm{F}_{Bearing}$, and $\mathrm{F}_{Contact}$ are the external, the bearing, and the contact forces, respectively. Components of mass and stiffness matrices are represented by Tables 1 and 2, respectively.

$$\left[\frac{1}{\Delta t^2}\mathbf{M}\right]\mathbf{u}_{t+\Delta t} = [\mathrm{F}_{External} + \mathrm{F}_{Bearing} + \mathrm{F}_{Contact}]_t - \left[\mathbf{K} - \frac{2}{\Delta t^2}\mathbf{M}\right]\mathbf{u}_t - \left[\mathbf{K} - \frac{2}{\Delta t^2}\mathbf{M}\right]\mathbf{u}_{t-\Delta t} \tag{19}$$

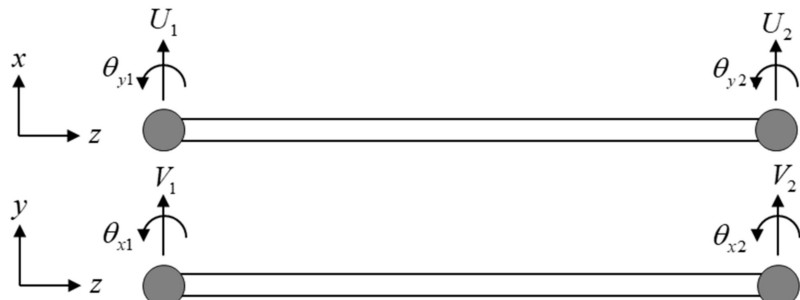

**Figure 3.** Coordinates of a beam element in xz and yz planes.

**Table 1.** Components of mass matrix.

| Symbol | Value |
|--------|-------|
| $M_1$ | $\frac{\rho AL}{840(1+\lambda)}\left(312 + 588\lambda + 280\lambda^2\right)$ |
| $M_2$ | $\frac{\rho AL^2}{840(1+\lambda)}\left(44 + 77\lambda + 35\lambda^2\right)$ |
| $M_3$ | $\frac{\rho AL}{840(1+\lambda)}\left(108 + 252\lambda + 140\lambda^2\right)$ |
| $M_4$ | $-\frac{\rho AL^2}{840(1+\lambda)}\left(26 + 63\lambda + 35\lambda^2\right)$ |
| $M_5$ | $\frac{\rho AL^3}{840(1+\lambda)}\left(8 + 14\lambda + 7\lambda^2\right)$ |
| $M_6$ | $-\frac{\rho AL^3}{840(1+\lambda)}\left(6 + 14\lambda + 7\lambda^2\right)$ |
| $M_7$ | $\frac{\rho I}{30(1+\lambda)^2 L}\left(36\right)$ |
| $M_8$ | $\frac{\rho I}{30(1+\lambda)^2 L}\left(3 - 15\lambda\right)$ |
| $M_9$ | $\frac{\rho IL}{30(1+\lambda)^2}\left(4 + 5\lambda + 10\lambda^2\right)$ |
| $M_{10}$ | $\frac{\rho IL}{30(1+\lambda)^2}\left(-1 - 5\lambda - 5\lambda^2\right)$ |

**Table 2.** Components of stiffness matrix.

| Symbol | Value |
|--------|-------|
| $K_1$ | $\frac{EI}{L^3(1+\lambda)}\left(12\right)$ |
| $K_2$ | $\frac{EI}{L^3(1+\lambda)}\left(6\right)$ |
| $K_3$ | $\frac{EI}{L^3(1+\lambda)}\left(4 + \lambda\right)$ |
| $K_4$ | $\frac{EI}{L^3(1+\lambda)}\left(2 - \lambda\right)$ |

The central differential method was used to numerically integrate the equation of motion, and Equation (19) is developed using an explicit method. The bearing reaction force and the contact force by the external force are calculated, and the shaft behavior is obtained by solving Equation (19). Then, the exact oil film thickness and bearing and contact forces generated at the next time step can be calculated. This procedure is repeatedly analyzed until the shaft behavior converges.

## 3. Numerical Calculation Procedure to Determine the Frictional Coefficients

The friction coefficient of journal bearings can be calculated by using Equation (12). The bearing and friction forces generated by the lubricant are determined according to the oil film thickness, which is always calculated even in very small oil film thickness. In this paper, the contact and friction forces calculated by surface roughness are only calculated in the area where oil film thickness is less than the four times of the standard deviation of the asperity heights ($4\sigma$) [20].

The proposed method is to calculate the frictional coefficients according to the rotational velocity to derive the Stribeck curve. Figure 4 is a flow chart to explain how to

derive the frictional coefficients. First, we set the design variables, operating conditions of the analysis model, and external forces. Then, the behavior of the shaft is calculated to determine the oil film thickness corresponding to the time step, and the oil film reaction and contact force are calculated at that time. Using these calculated forces, the friction coefficient and power loss are calculated with the method explained in Section 2.

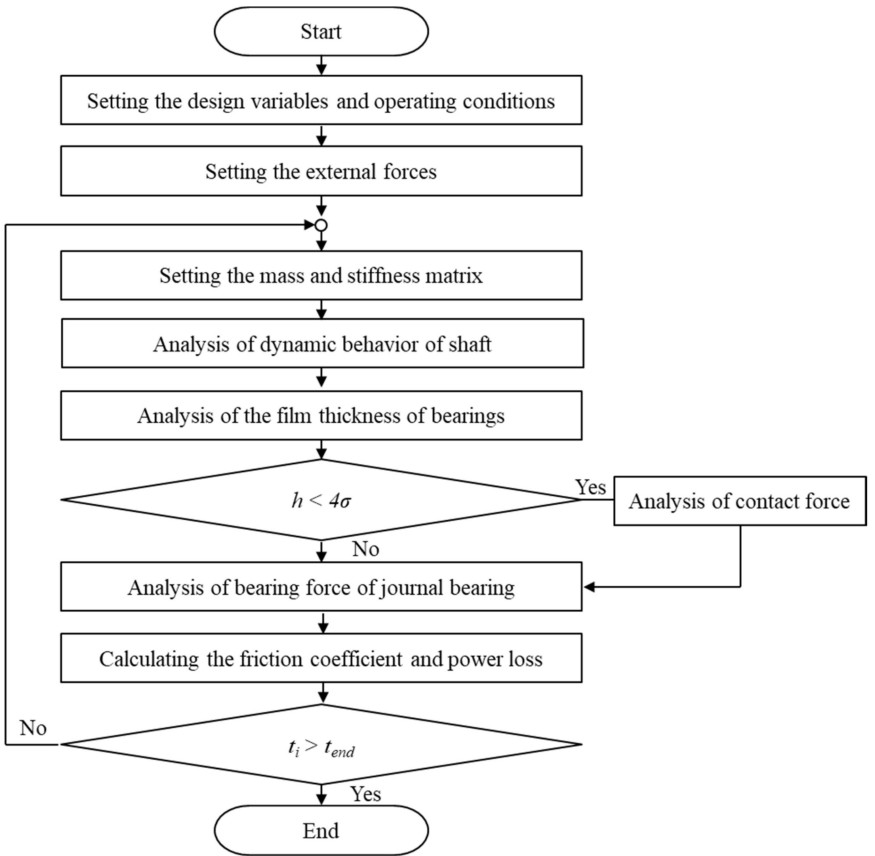

**Figure 4.** Analysis algorithm of friction coefficient and power loss.

## 4. Results and Discussion

### 4.1. Verification of the Proposed Method

To verify the proposed method in this research, we compared the calculated friction coefficients with the measured friction coefficients of journal bearings conducted by X. Lu et al. [6]. In the reference paper, an experimental device was constructed to measure the frictional coefficients of journal bearings according to various lubricants and two types of bearing materials. The journal bearing used in the experiment has a bearing length of 0.0254 m, a bearing radius of 0.012355 m, a clearance of 85 μm, shaft and bearing average asperity heights (Ra) of 0.32 μm, and an external force of 667 N is applied to the -x direction. Figure 5 shows the measured and calculated friction coefficients according to the rotating speed. It shows that the largest measured and calculated friction coefficients occur at 100 rpm. According to the numerical results, contact occurred in all rotational speed conditions. As the rotating speed increased, the friction coefficient was decreased due to increased bearing force. The maximum error between measured and calculated friction coefficients is about 20% at 100 rpm. At rotational speeds above 200 rpm, the error is less than 10%. In the experiment, the viscosity changes according to the operating speed because the temperature of the lubricant changes at each rotating speed. However, in our simulation, we used one nominal value of the viscosity of 0.0815 Pa·s in all rotating speed, which may result in the discrepancy between the experiment and simulation.

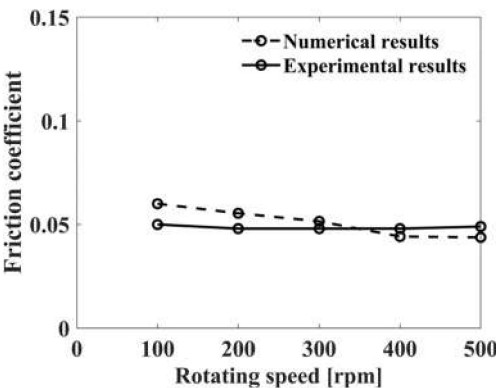

**Figure 5.** Comparison of simulated friction coefficients with measured ones.

### 4.2. Application of the Proposed Model to a Multiple-Stage Shaft Model of a SC

In this research, the multiple-stage shaft of a SC was used as an analytical model. The SC compresses internal gas by rotating a scroll existing at the top or bottom of the rotor. Figure 6 shows an analytical multiple-stage rotor model of a SC supported by two journal bearings (UB, LB). The parameters of the SC model, two journal bearings, and surface roughness of the rotational shaft and the journal bearings represented by Tables 3–5 respectively. The external forces are gas force, bearing forces at UB and LB, and centrifugal forces generated by the UBW (Upper Balance Weight) and LBW (Lower Balance Weight). The UBW and LBW are designed to compensate for the gas force through centrifugal forces. The masses of the UBW and LBW are 0.2224 kg and 0.0901 kg, respectively. The centrifugal forces according to the rotational speed are shown in Table 6. The gas force acting on the top of the SC is assumed to be $3000 + 250 \sin\theta$ N according to the rotational angle. The number of finite elements of a shaft to calculate the displacements and tilting angles is 50. The number of finite elements for the upper bearing and lower bearing to calculate oil film reaction forces and contact forces is 360, respectively. Furthermore, to calculate the friction coefficients and power loss according to the rotational speed, we performed the simulation from 20 to 200 Hz at intervals of 30 Hz.

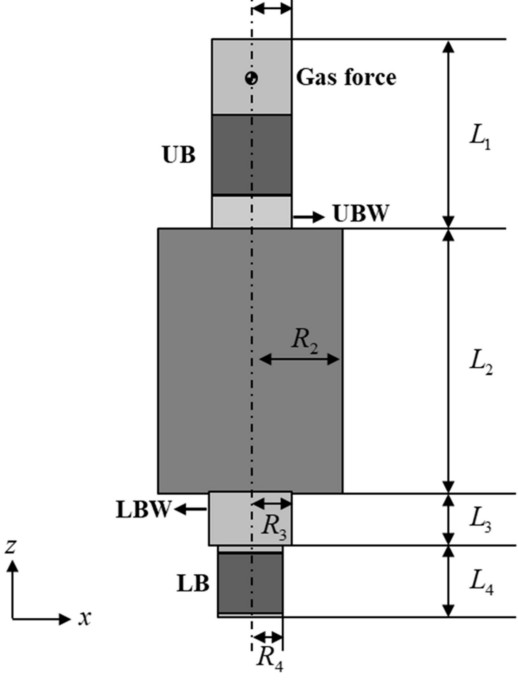

**Figure 6.** Scroll compressor (SC) model.

**Table 3.** Parameters of the SC model.

| Parameter | Value |
|---|---|
| $L_1$ | 80 mm |
| $L_2$ | 110 mm |
| $L_3$ | 27 mm |
| $L_4$ | 34 mm |
| $R_1$ | 17 mm |
| $R_2$ | 36 mm |
| $R_3$ | 26 mm |
| $R_4$ | 12.5 mm |

**Table 4.** Parameters of the journal bearings.

| Parameters | | Value |
|---|---|---|
| UB | Film thickness | 0.024 mm |
| | Distance from the SC bottom | 203.5 mm |
| LB | Film thickness | 0.02 mm |
| | Distance from the SC bottom | 16 mm |
| Viscosity | | $6 \times 10^{-3}$ Pa s |

**Table 5.** Parameters of surface roughness.

| Parameter | Value |
|---|---|
| $\sigma_{1,2}$ | 0.5 μm |
| $\beta_{1,2}$ | 6.5 μm |
| $\eta_{1,2}$ | $1.33 \times 10^{10}$ asp/m$^2$ |
| $E_{1,2}$ | 205 GPa |
| Shear modulus$_{1,2}$ | 102 GPa |
| Poisson's rate$_{1,2}$ | 0.3 |

**Table 6.** External forces of the balance weights according to operating speed.

| | Operating Speed | | | | | | |
|---|---|---|---|---|---|---|---|
| | 20 Hz | 50 Hz | 80 Hz | 110 Hz | 140 Hz | 170 Hz | 200 Hz |
| $F_{UBW}$ | 69.9 | 436.9 | 118.5 | 2114.8 | 3425.6 | 5051.0 | 6991.0 |
| $F_{LBW}$ | 22.9 | 142.9 | 365.9 | 691.9 | 1120.7 | 1652.5 | 2287.2 |

Using the proposed method, the friction coefficient according to the rotational speed was calculated. Figure 7 shows the simulated friction coefficients and the minimum film thickness of the UB and the LB. The contact between the shaft and the bearing only occurred at the UB because the gas force is applied to the upper part of the shaft close to upper bearing as shown in Figure 6. The simulated friction coefficient of the UB in Figure 7a shows that it increases quickly below the rotational speed of 80 Hz, with the smallest frictional coefficient when the solid contact begins to occur. The left region below 80 Hz can be regarded as mixed lubrication. In this region, there is contact in the UB because the bearing force is not sufficiently generated. The right region above 80 Hz can be regarded as hydrodynamic lubrication, and there is no solid contact in the bearing in that region. The minimum film thickness of the UB starts to decrease at a rotational speed above 140 Hz because the UBW near the bottom of the UB increased the centrifugal force. The LB is in the hydrodynamic lubrication regime with no solid contact. The simulated friction coefficient of the LB in Figure 7b shows that it increases with the increase of rotating speed below 110 Hz. However, in the operating conditions above 110 Hz, the increasing rate of the bearing force is almost proportional to the increasing rate of the friction force, which results in constant friction coefficients.

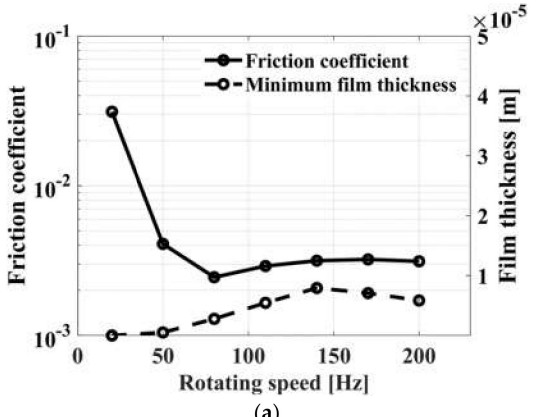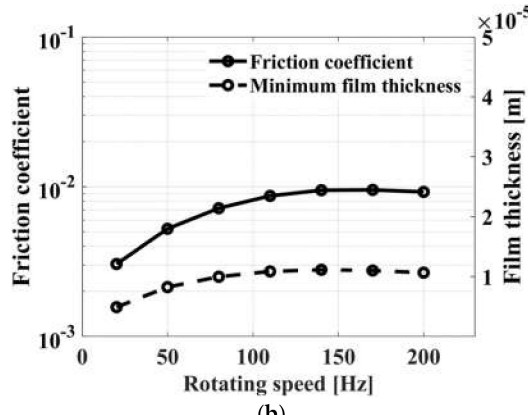

**Figure 7.** Friction coefficients of an upper bearing (**a**) and lower bearing (**b**) according to operating speed.

The friction coefficient in Figure 7 is the value using the mean friction coefficient in the last cycle of the analysis according to the rotating angle. The gas force of the simulated model changes according to the rotating angle, and therefore the friction coefficient changes for each rotating angle. The minimum, maximum, and average friction coefficients according to rotating angle were calculated and compared in Figure 8. The analysis results show that the maximum, minimum, and average friction coefficients of the UB and LB have similar trends.

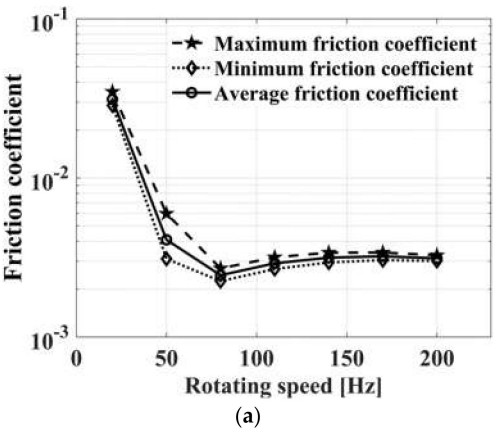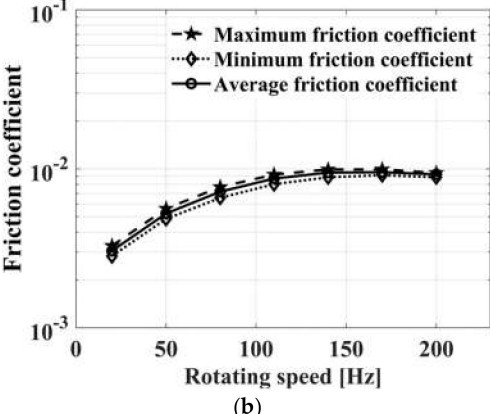

**Figure 8.** Friction coefficients of an upper bearing (**a**) and lower bearing (**b**) in case of maximum, minimum, and average.

The power loss of the UB and LB according to the rotational speed were calculated in Figure 9. The power loss of the UB in Figure 9a decreases until 80 Hz and increases with the increase of rotating speed. It can be explained that there is no solid contact above 80 Hz and that hydrodynamic power loss increases with the increase of rotating speed. Meanwhile, the power loss of the LB in Figure 9b increases as the rotating speed increases because the LB is only operated by hydrodynamic lubrication.

Figure 10 shows the total power loss in the bearing which is the summation of the asperity contact power loss and the hydrodynamic power loss. It shows that an asperity contact force exists below 80 Hz and it increases as the rotating speed decreases. In low rotating speeds, the hydrodynamic pressure is not sufficiently generated to balance the external forces, and the shaft contacts the bearing at the very small oil film thickness to generate asperity contact power loss. It also shows that hydrodynamic power loss only exists in the region above 80 Hz. It increases as the rotating speed increases.

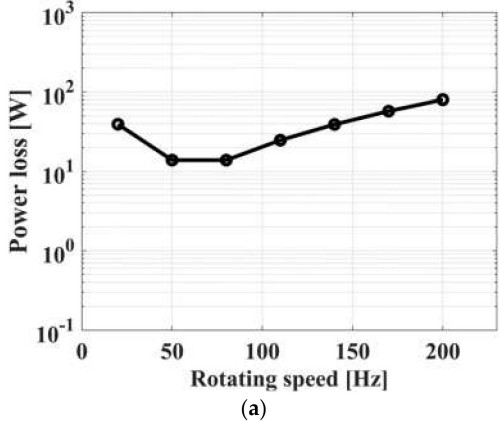
(**a**)

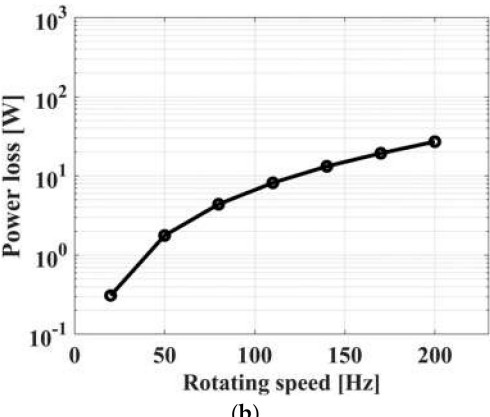
(**b**)

**Figure 9.** Power loss of an upper bearing (**a**) and lower bearing (**b**).

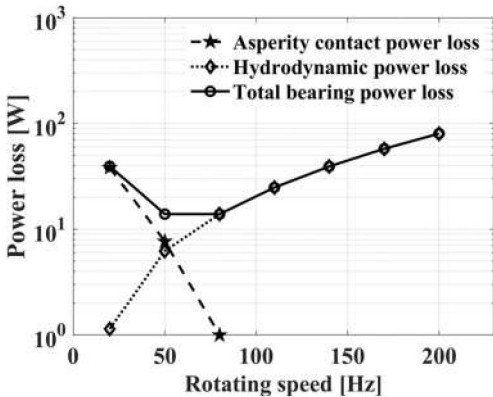

**Figure 10.** Asperity contact, hydrodynamic, and upper journal bearing power losses.

## 5. Conclusions

We propose a numerical method to derive the frictional characteristics of fluid film journal bearings considering the EHL and the asperity contact force. The dynamic analysis of a flexible shaft was solved by Timoshenko beam theory, and bearing pressure in the journal bearing considering shaft behavior was solved by using the average Reynolds equation with the FEM. When the contact occurred in the journal bearing, the asperity contact theory of Greenwood and Tripp was utilized to calculate the normal and tangential contact forces between two asperities of the shaft and bearing. Then, the proposed method was applied to investigate the friction coefficient and power loss for a rotor and bearing system of a SC. We can conclude as follows.

1.  We propose a method to determine the frictional coefficient of a journal bearing by utilizing contact forces of the asperity contact theory as well as the hydrodynamic bearing and friction forces of the lubricant. We also verified the proposed method with the experimental data of prior researchers.
2.  We apply the proposed method to a multiple-stage shaft model of a SC supported by two journal bearings and exerted by dynamic gas force and centrifugal force, and we investigated the frictional coefficients and the power loss at the mixed lubrication and hydrodynamic lubrication areas of the simulated model.
3.  The proposed method can be utilized to derive the friction coefficient and power loss of the SC according to the operating speed, and it can contribute to developing robust rotor and bearing systems supported by journal bearings.

**Author Contributions:** Conceptualization, G.H.; methodology, G.H.; software, G.H. and K.K.; validation, G.H.; formal analysis, G.H. and K.K.; investigation, G.H. and Y.P.; resources, G.H.; data curation, G.H.; writing—original draft preparation, G.H.; writing—review and editing, G.H. and G.J.; visualization, G.H.; supervision, G.J.; project administration, G.J.; funding acquisition, G.J. All authors have read and agreed to the published version of the manuscript.

**Funding:** This work was supported by the National Research Foundation of Korea (NRF) grant funded by the Korea government (MSIT) (No. 2020R1F1A1071308).

**Institutional Review Board Statement:** Not applicable.

**Informed Consent Statement:** Not applicable.

**Data Availability Statement:** Not applicable.

**Acknowledgments:** The authors acknowledge the funding support by the National Research Foundation of Korea (NRF) grant funded by the Korea government (MSIT) (No. 2020R1F1A1071308).

**Conflicts of Interest:** The authors declare no conflict of interest.

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
