# Peer review of "Numerical Determination of the Frictional Coefficients of a Fluid Film Journal Bearing Considering the Elastohydrodynamic Lubrication and the Asperity Contact Force"

_machines, doi:10.3390/machines10070494_

Round 1

Reviewer 1 Report

In this paper, a numerical method to derive the Stribeck curve considering the hydrodynamic force of a journal bearing and the asperity contact force of a rotating flexible shaft was proposed. The topic is interesting. However, there are some places that should be changed before the reviewer can recommend publication of the manuscript.
1.      Figures should be improved by using high definition version.
2.      In the section of “2.1 Reynolds equation considering EHL”, Is the Elastohydrodynamic Lubrication considered? if no, the title should be refined.
3.      The cavitation phenomenon is taken into account? If yes, what kind of the boundary condition is it?
4.      In fact, the temperature of oil film is important, as well as the viscosity. The effects of Viscosity-pressure and Viscosity-temperature are recommended to be taken into account.

Author Response

I appreciate your kind remarks and questions. They were very helpful to revise and to improve the manuscript. Please see the attached file.

Reviewer 2 Report

The authors should direct their efforts to prepare a manuscript that can bring some original contribution to the field of lubrication theory. In terms of fundamentals of fluid film lubrication, the results and analysis do not contribute to enlarge the understanding about lubrication regimes of cylindrical journal bearings. Moreover, some parts of the manuscript do not demonstrate a strong domain of knowledge about the basic concepts of fluid film bearings and finite element modeling of rotating systems.  In my opinion, the aim of the work must be changed to investigate the behavior of rotors of scroll compressors and not to determine the Stribeck curve. And the manuscript must be thoroughly reviewed to eliminate confused sentences and misconcepts and improve the description of the procedures employed to analyze the rotor-bearing system. All sections of the manuscript are in need of a thorough review. The authors must be very aware of their intended technical contribution to mechanical engineering. Please see attached file for more comments.

Author Response

(The authors gave the same response as above.)

Reviewer 3 Report

1. The conclusions indicate that a dynamic analysis of the flexible shaft has been performed. The trajectories of movement of the rotor points are the result of the dynamics calculation. However, no such results are presented in the work.

2. The thermal state of the bearings and the rotor as a whole is not specified in any way in the work.

3. Dynamic viscosity is denoted by the letter μ in equation (2). In this case, in equation (12), the same letter denotes the coefficient of friction.

4. The anti-wear properties of a lubricant have a significant effect on the Stribeck curve. How is this circumstance taken into account in the proposed methodology? It was necessary to indicate in the work a list of accepted assumptions.

Author Response

(The authors gave the same response as above.)

Round 2

Reviewer 2 Report

The authors have superficially attended to the mandatory changes recommended by the reviewer. Even the title of the manuscript is not proper. The reply about the shaft modeling is completely misleading. It is a strong misconcept to employ the Timoshenko beam theory in the finite element modeling neglecting the gyroscopic moments and the rotary inertia if the shear effects are taken into account. My opinion is that the current form of the manuscript is not ready to appear in an archival periodical in mechanical engineering.

It seems contradictory to consider this manuscript for publication in a special issue about Elastohydrodynamic Lubrication since the elastohydrodynamic effects are not the focus of the analysis presented.

Round 3

Reviewer 2 Report

All recommendations made by the reviewer have been attended by the authors.